# Non-linear trends using the generalized additive models in physical and technical performances of the Chinese Super League players

**Wei Zhang, Rancheng Tao, Fei Zhou, Changjing Zhou\*, Bo Gong**

School of Elite Sport, Shanghai University of Sport, Shanghai, China

\* zhouchangjing01@sus.edu.cn

## Abstract

This study aimed to investigate the non-linear evolution of position-specific physical and technical performance indicators across different tactical formations in the Chinese Super League (CSL) from the 2015 to the 2021 seasons. Match data were collected from 800 games played in six common formations (3-5-2, 4-3-3, 4-2-3-1, 4-4-2, 3-4-3, and 4-1-4-1). Players were classified into five positions: central defenders (CD; match observations = 2,219), fullbacks (FB; match observations = 2,060), central midfielders (CM; match observations = 2,786), wide midfielders (WM; match observations = 1,587), and forwards (FW; match observations = 1,680). Generalized additive models (GAMs) were constructed to analyze the nonlinear evolution trends in position-specific physical and technical performance across seasons and formations. The results showed that the total distance covered by CDs in the 3-5-2 formation showed a non-linear upward trend from the 2015 to 2021 seasons ($R^2$ = 0.3, $p$ = 0.006); FBs in the 3-4-3, 4-1-4-1, and 3-5-2 formations exhibited non-linear upward trends in number of sprints and sprint distance from the 2015 to 2021 seasons ($R^2$ = 0.189–0.448; $P$ < 0.03, respectively); the high-speed running of FWs in the 4-2-3-1 formation displayed a non-linear decline trend from the 2015 to 2021 seasons ($R^2$ = 0.477; $P$ < 0.001); CMs in the 4-2-3-1 formation demonstrated a non-linear increasing trend in the ball retention percentage ($R^2$ = 0.369; $p$ = 0.001); WMs in the 4-4-2 formation showed a non-linear decreasing trend in ball retention percentage and passes from the 2015 to 2021 seasons ($R^2$ = 0.259–0.29; $P$ = 0.006, respectively). These findings revealed non-linear evolutionary trends in physical and technical performances across positions and formations, providing an in-depth understanding of the changing match demands placed on players during matches. This enables the optimization of training and tactics by developing targeted strategies tailored to meet the specific requirements of different formations and playing positions.

## 1. Introduction

With the evolution of soccer matches, players' physical and technical performances have been continuously improving [1, 2]. Factors influencing players' performances include adjustments

**Data Availability Statement:** The datasets generated for this work have been uploaded as supplementary material.

**Funding:** The author(s) received no specific funding for this work.

**Competing interests:** The authors have declared that no competing interests exist.

in tactics and rules, increased application of technologies, enhanced professionalization, and improvements in players' physical and technical preparation [3–5]. Consequently, analyzing and evaluating variations in players' performances across different matches has become a crucial aspect of understanding the sport [6, 7]. Existing literature has revealed evolving trends in soccer matches from multiple perspectives based on different research objects [8, 9]. Studies examining the FIFA World Cup finals from 1966 to 2014 revealed a substantial rise of almost 35% in passes per minute, along with a 15% increase in game tempo [3]. These findings highlight the dynamic evolution of football at the global level over nearly five decades. Furthermore, an investigation conducted by Barnes et al. [8] illuminated significant changes in the English Premier League (EPL) during the 2006–2007 to 2012–2013 seasons, with remarkable 30–50% increases in high-intensity running and sprinting distances and 40% growth in passes. Yi et al. [10] demonstrated that throughout the nine seasons of the UEFA Champions League, while all defensive-related metrics remained stable, players performed an increasing number of passes per match, especially short passes.

However, soccer players in different playing positions require various physical capacities and technical abilities due to the different roles and tasks that they perform during the match [11–14]. Therefore, understanding each playing position's evolution in physical capacities and technical abilities is essential for player development, team composition, and strategic planning in soccer. Specifically, Peñas et al. [15] demonstrated that in the Spanish La Liga from the 2012/2013 to 2019/2020 season, central defenders experienced a significant increase in the total number of passes (30.6%), short passes (31.5%), and long passes (26.5%). In contrast, central midfielders (-7.2%), wide midfielders (-14.7%), and forwards (-11.3%) showed a significant decrease in their involvement in these technical actions. These findings suggest a shifting tactical approach where central defenders play an increasingly active role in build-up play. Similarly, research by Zhou et al. [16] on the Chinese Super League (CSL) reveals that from 2012 to 2017, there was a notable enhancement in high-intensity play and offensive actions, illustrating an increase in both physical and technical demands across different playing positions. Furthermore, Bush et al. [17] indicated that in the English Premier League over seven seasons (2006–07 to 2012–13), all playing positions showed an increase in high-intensity running, with fullbacks displaying the most significant rise of 36%. Similar trends were observed for sprint distance.

While the existing body of literature has extensively investigated the longitudinal trends in both physical and technical performance metrics among soccer players across various playing positions [15, 17], there is a noticeable gap in our understanding of how players have evolved within the context of different formations. It is imperative to delve into the non-linear evolution of physical and technical indicators among players deployed in distinct formations to gain a comprehensive insight into the multifaceted progress observed in contemporary soccer. This is particularly crucial because prior research has underscored that disparities in physical and technical performance among players occupying different positions are primarily influenced by their tactical roles within diverse formations [18–20]. Specifically, Forcher et al. [18] indicated that center backs covered the highest total and high-intensity distances in the 3-4-3 and 3-5-2 formations; wide midfielders showed more ball possessions, short, middle, and long passes, as well as fewer dribblings in the 4-4-2 diamond formation compared to other formations, whereas the number of dribbles for center backs and central midfielders remained consistent across formations.

Hence, the principal objective of this study pertains to the investigation of the nonlinear evolution of position-specific physical and technical parameters within the context of diverse tactical formations in the Chinese Super League (CSL), spanning from the 2015 to the 2021 seasons. By scrutinizing these developments over an extended temporal horizon, this research

endeavor aims to provide valuable insights into the dynamic progression of position-specific performance within the framework of various tactical formations.

## 2. Materials and methods

### 2.1. Participants and design

The participants in this study were elite football players from teams competing in the CSL. Data for this study were obtained from 1,364 matches played over the past seven consecutive seasons (2015–2021) of the CSL. Of these matches, 564 involved formation changes and were excluded. The remaining 800 matches without formation changes were included in the analysis. The starting formations used in these matches were as follows: 3-5-2 (n = 137), 4-3-3 (n = 77), 4-2-3-1 (n = 391), 4-4-2 (n = 257), 3-4-3 (n = 41), and 4-1-4-1 (n = 107). Player physical and technical performance indicators were collected for teams playing in 3-5-2, 4-4-2, 4-2-3-1, 4-1-4-1, 3-4-3, or 4-3-3 formations. Thus, the total sample comprised 10,332 individual match observations from 805 players.

The analysis only considered the results of players who played for at least half of the match, excluding goalkeepers due to the specificity of the playing position. Given that changes in positional roles can impact the player's performance (e.g., physical and technical indicators), the analysis excluded players who switched their tactical roles [20, 21]. Also, matches that included red cards were not analyzed. According to previous studies [18, 22], players were classified into five positions: central defenders (CD; match observations = 2,219), fullbacks (FB; match observations = 2,060), central midfielders (CM; match observations = 2,786), wide midfielders (WM; match observations = 1,587), and forwards (FW; match observations = 1,680). In the 3-4-3 formation (CD; n = 116, FB; n = 87, CM; n = 80, and FW; n = 120), the 3-5-2 formation (CD; n = 419, FB; n = 282, CM; n = 426, and FW; n = 286), the 4-3-3 formation (CD; n = 157, FB; n = 156, CM; n = 237, and FW; n = 225), the 4-4-2 formation (CD; n = 520, FB; n = 524, WM; n = 532, CM; n = 524, and FW; n = 526), the 4-1-4-1 formation (CD; n = 218, FB; n = 216, CM; n = 329, WM; n = 216, and FW; n = 113), and the 4-2-3-1 formation (CD; n = 789, FB; n = 795, CM; n = 1190, WM; n = 820, and FW; n = 410), as shown in Table 1. The study design and procedures were in accordance with the Declaration of Helsinki and approved by the ethics committee at Shanghai University of Sport. This study utilized publicly available match data for analysis and did not involve the collection of players' personal information. Therefore, informed consent was not required for this study.

### 2.2. Procedure

The analysis only considered teams that maintained a consistent formation throughout the entire match, as suggested previously [18, 20, 23]. In two stages, the consistency of team

**Table 1. Descriptive statistics of player data for each position in different formations.**

| Formation | CD | FB | CM | WM | FW |
|-----------|-----|-----|------|-----|-----|
| 3-4-3 | 116 | 87 | 80 | | 120 |
| 3-5-2 | 419 | 282 | 426 | | 286 |
| 4-3-3 | 157 | 156 | 237 | | 225 |
| 4-4-2 | 520 | 524 | 524 | 532 | 526 |
| 4-1-4-1 | 218 | 216 | 329 | 216 | 113 |
| 4-2-3-1 | 789 | 795 | 1190 | 820 | 410 |

CD = central defender; FB = fullback; CM = central midfielder; WM = wide midfielder; FW = forward.

**Table 2. Technical and physical performance-related parameters.**

| Physical Performance-Related Parameters: Operational Definition |
| --- |
| **Distance covered (km)**: Distance covered in a match. |
| **Low speed running**: Distance covered at a speed of 7.1->14.3km/h in a match. |
| **Middle speed running**: Distance covered at a speed of 14.3->19.7km/h in a match. |
| **High speed running**: Distance covered at a speed of 19.7->25.1km/h in a match. |
| **Sprint**: Distance covered at a speed over 25.1km/h in a match. |
| **Number of sprints**: Number of sprints covered at the speed over 25.1km/h in a match. |
| **Distance covered ball in play (km)**: Distance covered when the ball was in play. |
| **Technical Performance-Related Parameters: Operational Definition** |
| **Pass**: an intentional played ball from one player to another. |
| **Pass success rate**: successful passes as a proportion of total passes. |
| **Challenge**: actions when two players are competing for ball possession, which is not in the control of any player, i.e., both players have approximately a 50% chance of gaining control of the ball; includes ground and air challenges. |
| **Challenge success rate**: successful challenges as a proportion of the total challenges. |
| **Ball retention percentage**: ball retention percentage refers to the percentage of time a player to keep possession of the ball during a match. |
| **Gain of possession**: the action of gaining possession from an opposition player who is in possession of the ball. |
| **Loss of possession**: the moment when a player who had control of the ball loses it to an opponent, either through a mistake or due to the opponent's skillful intervention. |

formations was examined. In the first stage, the tactical formation for each team and match was determined using official match reports from the CSL, which were provided by Amisco Pro® (Amisco, Nice, France). The accuracy, validity, and reliability of Amisco Pro®'s working process have been thoroughly discussed in prior studies [24, 25]. In the second stage, we conducted a secondary verification of the average player positioning reported in official match reports against the match formations provided by a publicly accessible football statistics website known as "whoscored.com" (http://www.whoscored.com), whose original data from the OPTA Sportsdata company has been tested with a respectable level of inter-operator reliability (Kappa values > 0.90) [26].

The physical and technical indicators of players in different positions across various formations are collected by a semi-automatic computerized video tracking system, Amisco Pro® (Amisco, Nice, France). The physical variables included the following: total distance covered (km), distance covered ball in play (km), number of sprints, sprint (25.1->km/h), high speed running (19.7->25.1km/h), middle speed running (14.3->19.7km/h), and low speed running (7.1->14.3km/h). Additionally, the technical variables included are the following: gain of possession, loss of possession, ball retention percentage, challenge, challenge success rate, passes, and pass success rate. The operational definitions of technical and physical performance-related parameters are presented in Table 2.

## 2.3. Statistical analysis

Data analysis was performed using RStudio (Version 4.0.2) and R software (Version 4.3.0). In order to flexibly model potential nonlinear relationships between tactical formations and players' performance, generalized additive models (GAMs) were constructed using the gamm4 package in R. GAMs avoid predefining rigid parametric functions and can estimate smooth regression curves directly from the data [27]. To take into account individual variability and correlations between the repeated measures data of the same player under different formations, players were included as a random effect in the GAMs. The physical and technical

indicators were separately modeled as dependent variables, with tactical formation as the independent variable. GAM plotting functions were used to visualize the estimated smooth regression curves, demonstrating position-specific patterns in the different formations. Statistical significance was defined as $P<0.05$.

## 3. Result

As shown in Fig 1, the total distance covered by CDs in the 3-5-2 formation displayed a non-linear upward trend from the 2015 to 2021 seasons ($R^2 = 0.3$; $P = 0.006$). Furthermore, the sprint distance and number of sprints completed by CDs in the 4-4-2, 4-2-3-1, 3-5-2, and 3-4-3 formations displayed non-linear upward trends from the 2015 to 2021 seasons ($R^2 = 0.143$–0.232; $P< 0.04$, respectively). Fig 2 demonstrated that in the 3-4-3, 4-1-4-1, and 3-5-2 formations, FBs exhibited non-linear upward trends in number of sprints and sprint distance from the 2015 to 2021 seasons ($R^2 = 0.189$–0.448; $P < 0.03$, respectively). For CMs, Fig 3 illustrated that the total distance covered in the 3-4-3 and 3-5-2 formations displayed non-linear upward trends from the 2015 to 2021 seasons ($R^2 = 0.276$–0.616; $P < 0.03$, respectively). Moreover, the sprint distance and high-speed running of CMs in the 4-4-2 formation exhibited nonlinearly increasing trends ($R^2 = 0.253$–0.309; $P < 0.04$, respectively). CMs in the 3-5-2 formation demonstrated non-linear increases in high-speed running, number of sprints, and sprint distance from the 2015 to 2021 seasons ($R^2 = 0.541$–0.567; $P < 0.001$, respectively). In addition, Fig 4 showed that the distance covered ball in play by WMs in the 4-1-4-1 formation exhibited a non-linear downward trend from the 2015 to 2021 seasons ($R^2 = 0.359$; $P = 0.04$), while their counterparts in the 4-2-3-1 formation showed a non-linear upward trend ($R^2 = 0.334$; $P < 0.001$). Finally, the physical performance of FWs also exhibited notable trends. Specifically, Fig 5 indicated that the high-speed running of FWs in the 4-2-3-1 formation displayed a non-linear decline trend from the 2015 to 2021 seasons ($R^2 = 0.477$; $P < 0.001$). In contrast, the high-speed running of FWs in the 4-4-2 formation showed a non-linear increase trend from the 2015 to 2021 seasons ($R^2 = 0.513$; $P = 0.04$).

Concerning technical performance, the passes made by CDs in formations such as 4-2-3-1, 4-4-2, 3-4-3, 4-1-4-1, 4-3-3, and 3-5-2 exhibited non-linear upward trends between the 2015 and 2021 seasons ($R^2 = 0.233$–0.377; $P < 0.009$, respectively), as shown in Fig 1. At the same time, the challenges performed by FBs in the 4-2-3-1 and 3-5-2 formations exhibited non-linear decreasing trends from the 2015 to 2021 seasons ($R^2 = 0.0815$–0.117; $P < 0.001$, respectively), as depicted in Fig 2. Additionally, FBs in the 4-2-3-1 formation saw a non-linear increasing trend in ball retention percentage ($R^2 = 0.158$; $P < 0.001$), while those in the 3-5-2 and 3-4-3 formations experienced non-linear decreasing trends over the 2015–2021 seasons ($R^2 = 0.237$–0.637; $p = 0.03$, respectively). Regarding CMs, Fig 3 demonstrated that from the 2015 to 2021 seasons, the ball retention percentage of CMs in the 4-2-3-1 formation demonstrated a non-linear increasing trend ($R^2 = 0.369$; $p = 0.001$). In contrast, the ball retention percentage of their counterparts in the 3-5-2 and 3-4-3 formations showed non-linear decreasing trends ($R^2 = 0.165$–0.566; $p < 0.05$, respectively). Fig 4 illustrates that WMs in the 4-4-2 formation showed a non-linear decreasing trend in ball retention percentage and passes from the 2015 to 2021 seasons ($R^2 = 0.259$–0.29; $P = 0.006$, respectively). Finally, the challenge success rate of FWs in the 4-2-3-1 formation exhibited a non-linear increasing trend between the 2015 and 2021 seasons ($R^2 = 0.098$; $p = 0.005$), as shown in Fig 5.

## 4. Discussion

The main finding of this study is that the physical and technical performance of players in different positions shows different non-linear evolution trends across different formations. This

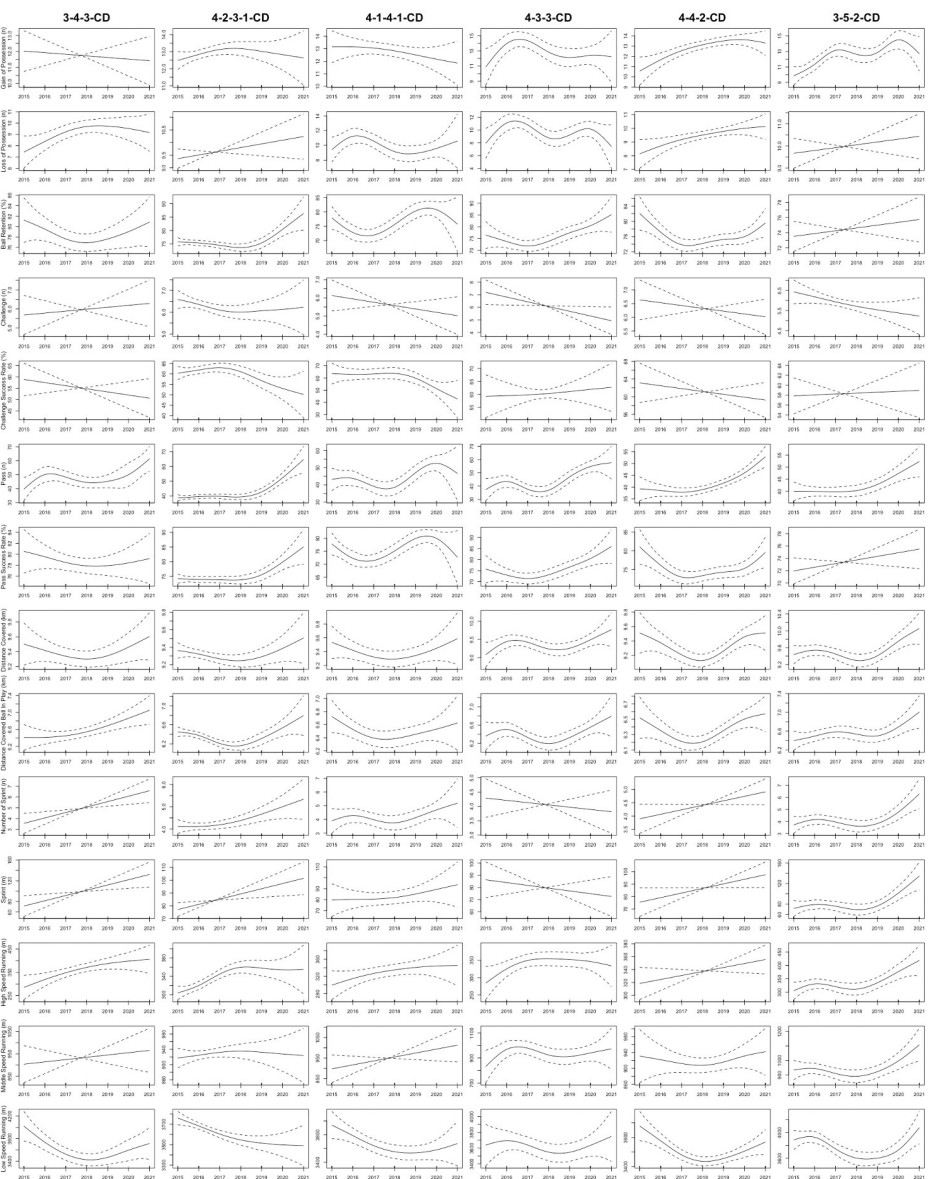

**Fig 1. The non-linear evolution trends of the physical and technical performance of CDs in the different formations from the 2015 to 2021 seasons.** Note: 3-4-3-CD = central defenders in the 3-4-3 formation; 4-2-3-1-CD = central defenders in the 4-2-3-1 formation; 4-1-4-1-CD = central defenders in the 4-1-4-1 formation; 4-3-3-CD = central defenders in the 4-3-3 formation; 4-2-2-CD = central defenders in the 4-4-2 formation; 3-5-2-CD = central defenders in the 3-5-2 formation. Solid lines indicate smoothers surrounded by 95% confidence intervals depicted by dashed lines.

suggests that the evolution of the physical and technical indicators of players depends to a large extent on their positions and tactical formations [9, 12, 15, 20].

## 4.1. Central defender

Previous research on the English Premier League from the 2006–07 to 2012–13 seasons has revealed a significant 70% increase in the number of passes made by CDs [17]. Consistent with these findings, our study also indicated non-linear upward trends in the number of passes

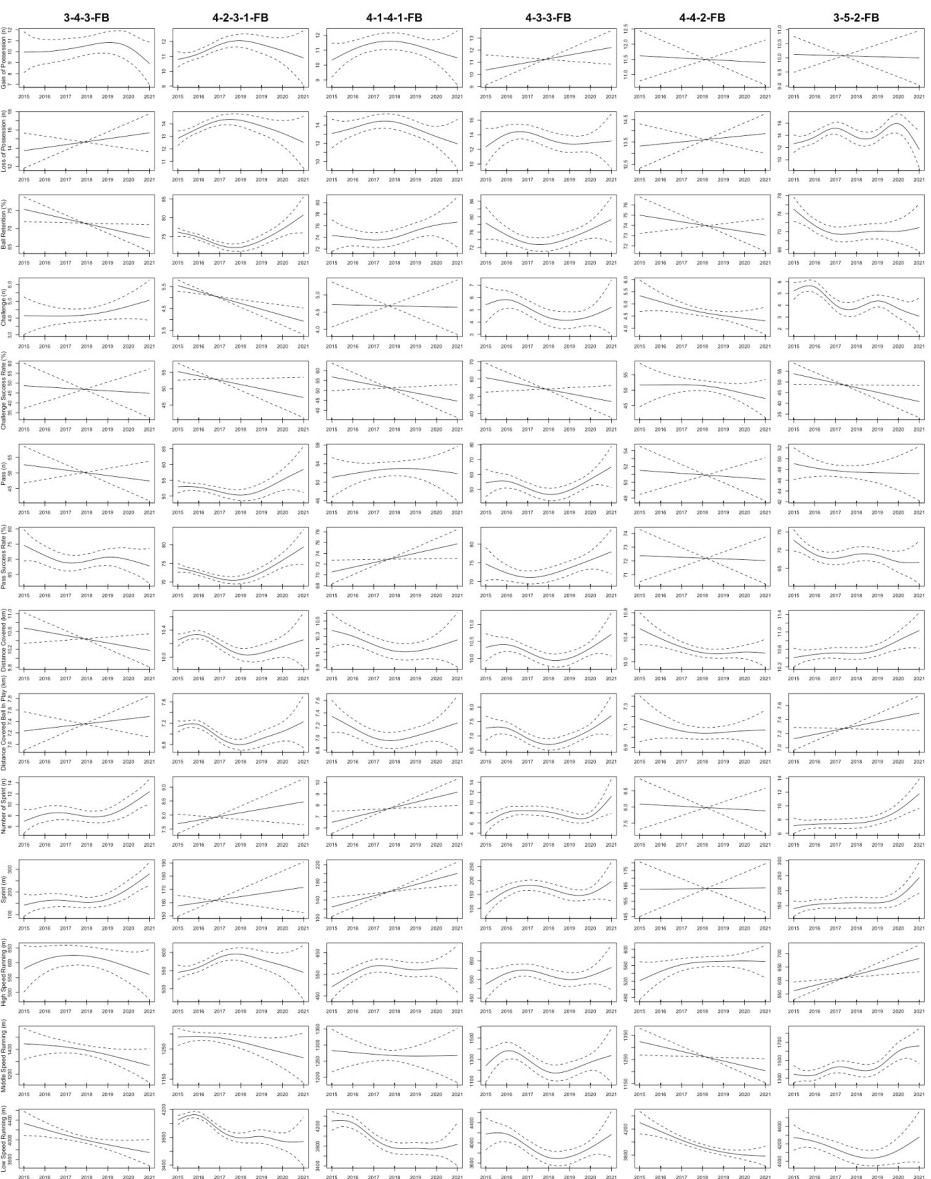

**Fig 2. The non-linear evolution trends of the physical and technical performance of FBs in the different formations from the 2015 to 2021 seasons.** Note: 3-4-3-FB = fullbacks in the 3-4-3 formation; 4-2-3-1-FB = fullbacks in the 4-2-3-1 formation; 4-1-4-1-FB = fullbacks in the 4-1-4-1 formation; 4-3-3-FB = fullbacks in the 4-3-3 formation; 4-2-2-FB = fullbacks in the 4-4-2 formation; 3-5-2-FB = fullbacks in the 3-5-2 formation. Solid lines indicate smoothers surrounded by 95% confidence intervals depicted by dashed lines.

completed by CDs across all formations between the 2015 and 2021 seasons. This trend suggests that the role of CDs has evolved to involve more participation in build-up play and distribution of the ball from the backline [15, 17, 28]. Additionally, the CDs in the 4-2-3-1, 3-5-2, and 4-3-3 formations showed non-linear upward trends in ball retention percentage and pass success rate. This result may demonstrate that compared to the other three formations (such as 4-1-4-1, 3-4-3, and 4-4-2), the CDs in the 4-2-3-1, 3-5-2, and 4-3-3 formations require stronger ball control ability and passing accuracy. Furthermore, in the 4-1-4-1, 4-4-2, 3-5-2, 4-3-3, and 4-2-3-1 formations, the challenges made by CDs showed non-linear downward trends

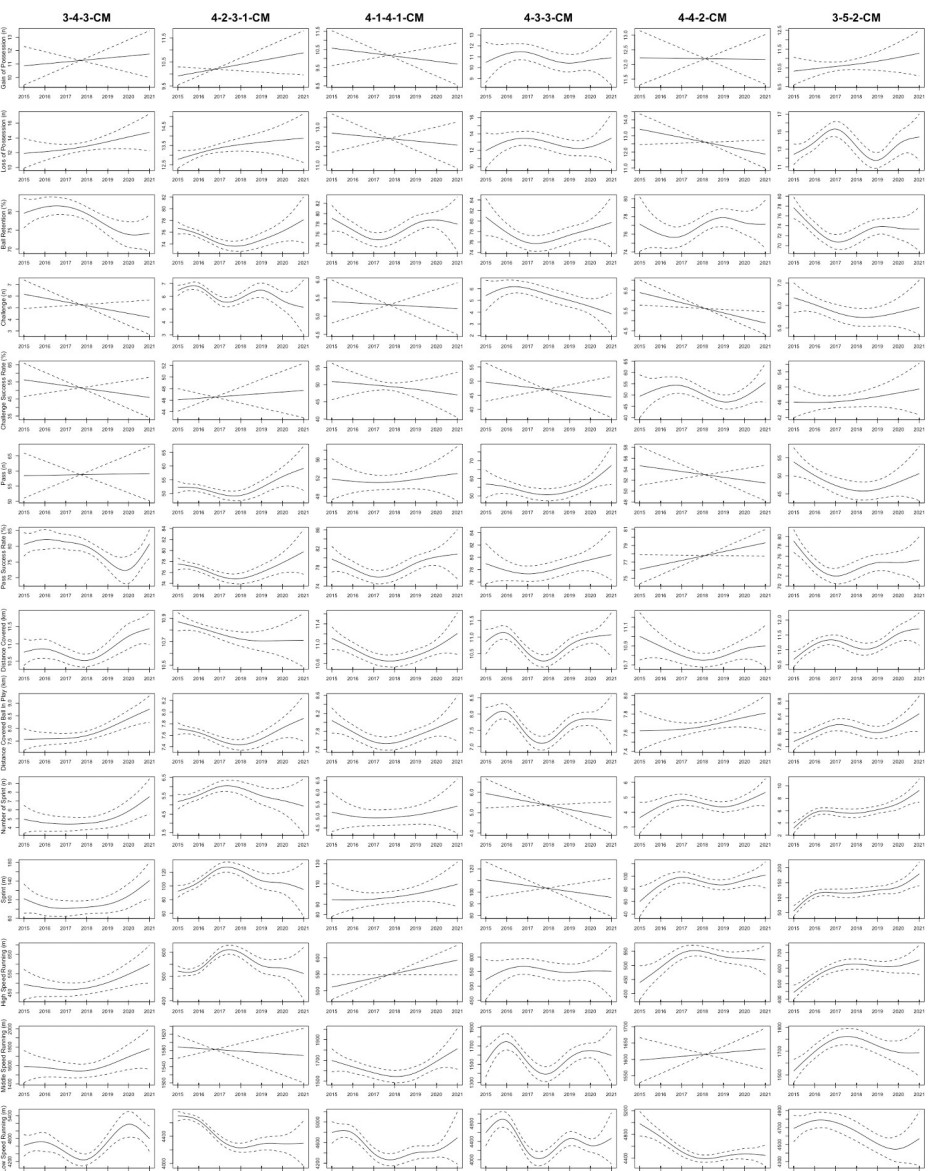

**Fig 3. The non-linear evolution trends of the physical and technical performance of CMs in the different formations from the 2015 to 2021 seasons.** Note: 3-4-3-CM = central midfielders in the 3-4-3 formation; 4-2-3-1-CM = central midfielders in the 4-2-3-1 formation; 4-1-4-1-CM = central midfielders in the 4-1-4-1 formation; 4-3-3-CM = central midfielders in the 4-3-3 formation; 4-2-2-CM = central midfielders in the 4-4-2 formation; 3-5-2-CM = central midfielders in the 3-5-2 formation. Solid lines indicate smoothers surrounded by 95% confidence intervals depicted by dashed lines.

from the 2015 to 2021 seasons. These results may suggest that players play soccer with more tactical discipline and less aggressiveness [29, 30]. However, the CDs in the 3-4-3 formation have shown an upward trend in terms of challenges. This may be due to the fact that in the 3-4-3 formation, FBs can be more offensive [18], which puts greater defensive responsibilities on the CDs, requiring them to engage in more challenges to disrupt opposition attacks and regain possession.

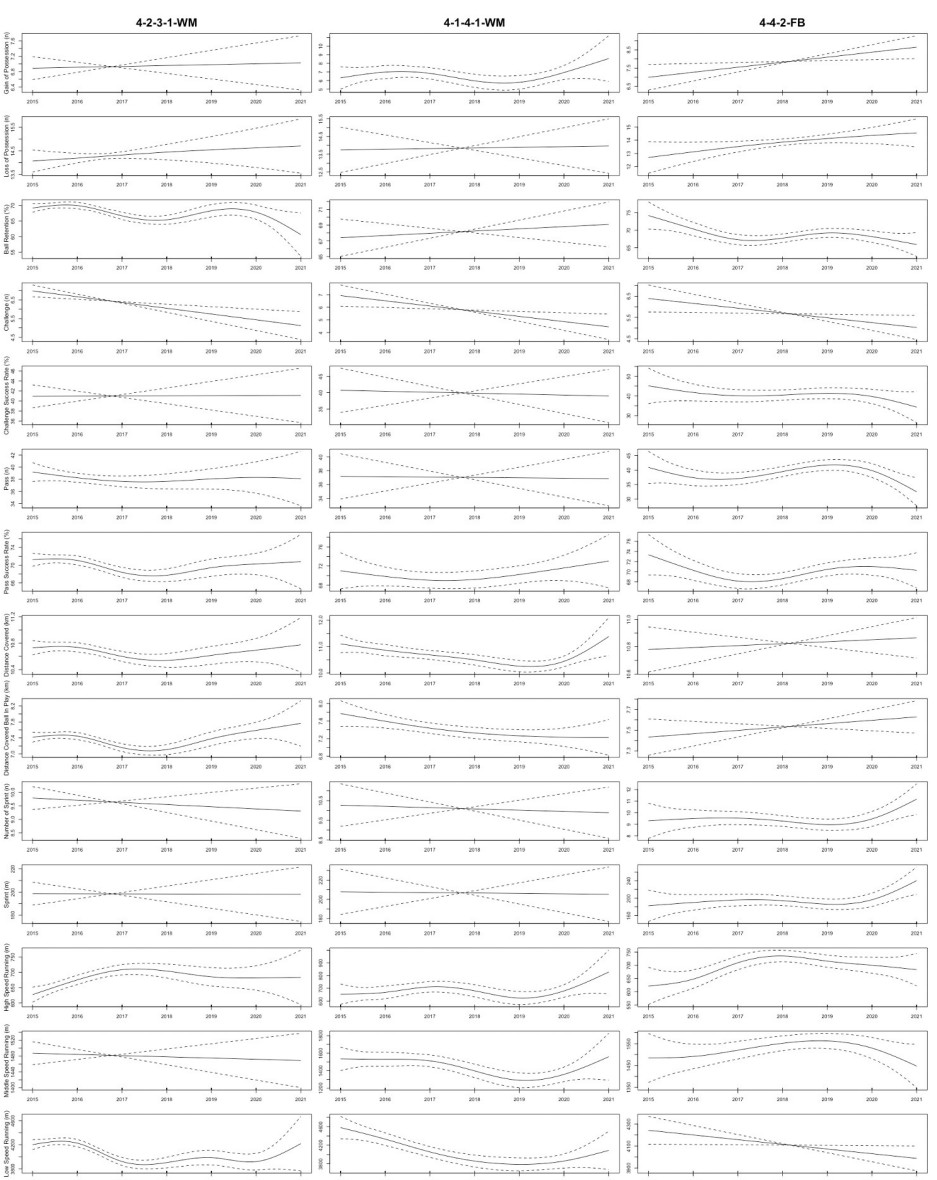

**Fig 4. The non-linear evolution trends of the physical and technical performance of WMs in the different formations from the 2015 to 2021 seasons.** Note: 4-2-3-1-WM = wide midfielders in the 4-2-3-1 formation; 4-1-4-1-WM = wide midfielders in the 4-1-4-1 formation; 4-2-2-WM = wide midfielders in the 4-4-2 formation. Solid lines indicate smoothers surrounded by 95% confidence intervals depicted by dashed lines.

Previous research on the Second Division of Japanese Professional Football indicates that the total running distance of players has shown a decreasing trend over three seasons [31]. However, our analysis showed that from 2015 to the 2021 seasons, the total distance covered by CDs in formations like 4-2-3-1, 4-3-3, 3-4-3, and 3-5-2 demonstrated nonlinear growth trends. The increased distance covered by CDs reflects the evolving tactical demand for defenders to be more involved in building up plays from the back. With more teams adopting possession-based playing styles [32], CDs are expected to push higher up the pitch and link up with midfielders, which leads to greater distance covered. Furthermore, the CDs in the 4-4-2,

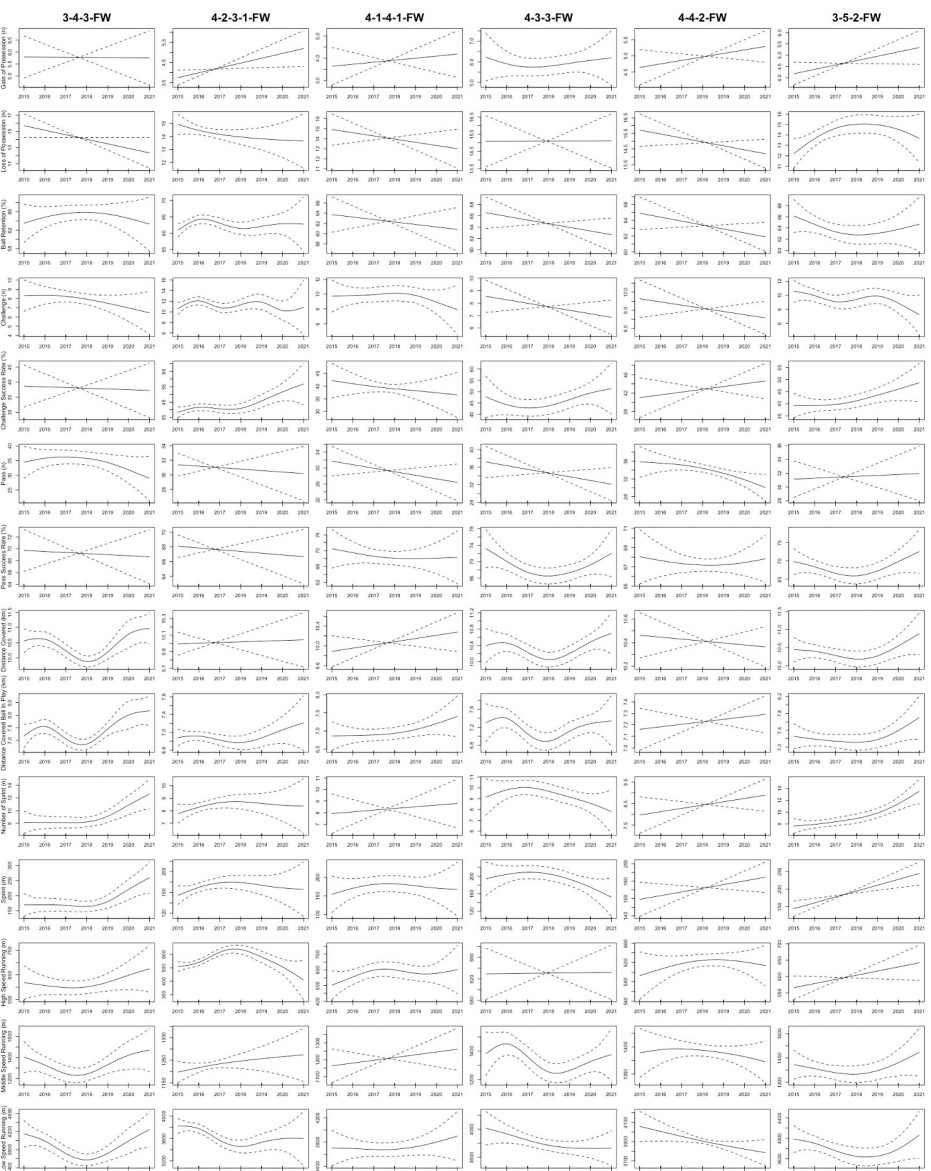

**Fig 5. The non-linear evolution trends of the physical and technical performance of FWs in the different formations from the 2015 to 2021 seasons.** Note: 3-4-3-FW = forwards in the 3-4-3 formation; 4-2-3-1-FW = forwards in the 4-2-3-1 formation; 4-1-4-1-FW = forwards in the 4-1-4-1 formation; 4-3-3-FW = forwards in the 4-3-3 formation; 4-2-2-FW = forwards in the 4-4-2 formation; 3-5-2-FW = forwards in the 3-5-2 formation. Solid lines indicate smoothers surrounded by 95% confidence intervals depicted by dashed lines.

4-1-4-1, 4-2-3-1, 3-4-3, and 3-5-2 formations have exhibited nonlinear upward trends in the number of sprints and sprint distance from the 2015 to 2021 seasons. This result may demonstrate that the high-intensity demands imposed on players have increased over time [8, 15]. However, the CDs in the 4-3-3 formations showed the opposite trend. A plausible explanation for this result could be that the 4-3-3 formation places a significant emphasis on midfield control and possession play [33]. With the presence of three CMs, there is already a higher number of players occupying the attack zone, which reduces the necessity for the CDs to push forward. Their primary responsibility lies in preserving a robust defensive system and preventing the

opposition's attempts to penetrate through the middle. Consequently, their involvement in high-intensity running becomes less crucial as they prioritize defensive stability over offensive contributions.

## 4.2. Fullback

Regarding technical performance, the FBs in the 3-4-3 and 3-5-2 formations have exhibited non-linear downward trends in the number of passes, pass success rate, and ball retention percentage from the 2015 to 2021 seasons. This finding is not surprising, as previous studies have demonstrated that the 3-4-3 and 3-5-2 formations impose greater physical demands on the FBs [18, 20, 34]. Consequently, it can be inferred that the decrease in ball retention percentage and passes could potentially be compensated by an increase in running output, such as the number of sprints and the sprint distance. In contrast, FBs in the 4-2-3-1 formation showed a non-linear increase in these technical metrics over time. This can be attributed to the tactical structure of the formation [20]. The presence of two CMs in a double pivot provides additional support and passing options for the FBs. This allows them to make more accurate passes while maintaining possession. Furthermore, FBs in the 4-4-2, 4-2-3-1, 4-3-3, and 3-5-2 formations have exhibited a nonlinear downward trend in the number of challenges between the 2015 and 2021 seasons. The diminishing challenge faced by FBs, similar to what has been observed among CDs, might indicate a shift towards a less aggressive style of play among players [29, 30].

Match analysis has shown that high-intensity actions play a crucial role in football, predominantly due to their association with training status and goal-scoring situations, as well as their ability to distinguish between different levels of play [11, 15, 35]. According to Zhou et al. [36], the increase in high-intensity running indicates an increase in the level of competition in the CSL. Our data demonstrated that the number of sprints and sprint distances covered by FBs in the 4-1-4-1, 4-2-3-1, 4-3-3, 3-4-3, and 3-5-2 formations have shown non-linear upward trends from the 2015 to 2021 seasons. These findings highlighted the increasing high-intensity demands for FBs in different formations. A reasonable explanation for this finding could be that FBs are no longer solely defensive players but have become key contributors in both defensive and offensive phases [37]. They are often involved in pressing high up the pitch, supporting the midfield, and providing an extra attacking option. As a result, FBs may perform their offensive and defensive tasks with more sprinting output.

## 4.3. Central midfielder

According to our data, CMs in the 4-2-3-1 formation have shown nonlinear growth trends in the number of passes and ball retention percentage from the 2015 to 2021 seasons. This is in line with the findings of Aquino et al. [38], who observed that teams employing a 4-2-3-1 formation tended to dominate possession and displayed enhanced network properties, suggesting better team coordination and player interaction. Furthermore, it is widely acknowledged that CMs play a crucial role in coordinating offensive actions [34, 39]. Thus, this trend may also be attributed to build-up play, which allows CMs in the 4-2-3-1 formation to establish a connection between the defensive line and the forward line through passing. Furthermore, previous studies have indicated significant differences in player performance between 3-back formations (e.g., 3-5-2) and 4-back formations (e.g., 4-4-2) [18, 34]. In line with these findings, we discovered a nonlinear increase in pass success rate for CMs in the 4-2-3-1, 4-4-2, and 4-3-3 formations between the 2015 and 2021 seasons. The tactical evolution in modern football has emphasized the importance of CMs as playmakers [7], resulting in improved passing accuracy. However, the pass success rate performed by CMs in the 3-5-2 formation has demonstrated a

nonlinear decline trend. This could be attributed to the emphasis on wing attacks in the 3-5-2 formations. Additionally, in the 3-5-2 formation, the CDs often play a crucial role in orchestrating the team's attack [39]. Consequently, the pass success rate of CMs experiences a decline over time.

A previous study reported a significant reduction in the total distance covered by CMs in the Spanish La Liga over eight seasons [15]. However, our study's findings contradict this trend. We observed a non-linear growth pattern in the total distance covered by CMs in various formations such as 4-1-4-1, 4-3-3, 3-5-2, and 3-4-3 between the 2015 and 2021 seasons. This result can be attributed to the evolving roles of CMs, particularly the increasing emphasis on box-to-box midfielders who contribute both defensively and offensively. Furthermore, high-intensity running has been used to represent the physical demands of soccer match play [11, 40]. because it correlates well with physical capacity [41] and differentiates between competitive levels and genders [42, 43]. In our study, we found that CMs in formations such as 4-1-4-1, 4-4-2, 3-4-3, and 3-5-2 exhibited non-linear increasing trends in terms of high-speed running, sprinting, and number of sprints from the 2015 to 2021 seasons. As aforementioned, with the evolving roles of CMs (e.g., box-to-box midfielders), they are expected to contribute significantly to both the offensive and defensive aspects of their teams. As a result, they are required to perform more high-intensity running to fulfill their offensive and defensive responsibilities effectively. However, CMs in the 4-3-3 formation have exhibited a linear decrease in both sprint distance and the number of sprints between the 2015 and 2021 seasons. Previous studies have concluded that the decline in running performance is possibly due to superior passing performance in the CSL [36]. In detail, the 4-3-3 formation places a significant emphasis on controlling the midfield and maintaining possession [33]. As a result, the demand for constant sprinting from CMs may have decreased.

## 4.4. Wide midfielder

According to recent research, the physical demands and technical-tactical activities performed by professional soccer players are strongly dependent on the tactical formation used [34, 44, 45]. We observed that the performance of WMs in different formations has exhibited interesting trends. Specifically, WMs in the 4-1-4-1 formation have shown a non-linear upward trend in pass success rate and ball retention percentage from the 2015 to 2021 seasons. This suggests that these players have become more effective at maintaining possession and executing successful passes in their roles. Conversely, WMs in the 4-4-2 formation have experienced a decline in these metrics during the same period. These results could reflect changes in the duties of the different positional roles [46]. Specifically, compared to their counterparts in the 4-1-4-1 formation, WMs may have greater defensive responsibilities in the 4-4-2 formation [47]. They may need to track back more frequently to provide defensive cover or mark opposing FBs. This defensive focus can limit their opportunities to participate in the build-up play and subsequently reduce their passing involvement and success rate.

In terms of physical performance, WMs in the 4-1-4-1 formation have shown a downward trend in distance covered ball in play, sprinting, and number of sprints from the 2015 to 2021 seasons. We presume that this result is related to the superior passing performance of WMs in the 4-1-4-1 formation. Furthermore, previous studies have shown that the WM is a position that requires high physical capacity [48]. We have discovered that from the 2015 to 2021 seasons, the WMs in the 4-4-2 formation exhibited a non-linear upward trend in total distance covered, sprinting, number of sprints, and high-speed running. González Rodenas et al. [47] stated that WMs have a more defensive role in the 4-4-2 formation. Therefore, the WMs in the

4-4-2 formation may need to fulfill their defensive duties (e.g., providing defensive protection and pressing the opponent quickly) by making more running outputs.

### 4.5. Forward

Regarding the technical performance of FWs, Bush et al. [46] demonstrated a 25% increase in passes made by FWs in the English Premier League over seven seasons. However, Peñas et al. [15] stated a significant reduction in the number of passes executed by FWs in the Spanish La Liga across eight seasons. One limitation of both studies is their exclusive consideration of player positions, without accounting for the impact of tactical formations on player performance. It is essential to recognize that players' performances can exhibit different trends depending on the formations they play in. Specifically, it was observed in our study that FWs in the 3-5-2 formation exhibited a non-linear upward trend in pass and pass success rates from the 2015 to 2021 seasons. However, FWs in the 4-2-3-1, 4-4-2, 4-3-3, 4-1-4-1, and 3-4-3 formations showed a non-linear declining trend in the same aspects during the same period. As the game evolves, defenses have become more organized and compact [49], which makes it more difficult for FWs to execute accurate passes and maintain a high passing success rate. Additionally, increased focus on pressing and defensive tactics may limit the time and space available for FWs to execute passes.

The observed trend in this study is that the total running distance of FWs in the 4-1-4-1, 4-2-3-1, 4-3-3, 3-4-3, and 3-5-2 formations showed non-linear increase trends during the 2015–2021 seasons. This could be attributed to improvements in players' physical preparation [8]. However, it has been reported that the distance covered at high intensity seems to be a superior, more sensitive performance indicator than the total distance covered, as it correlates strongly with physical capacity [17, 50, 51]. Specifically, Bush et al. [17] demonstrated a significant increase in high-intensity running among FWs during seven seasons in the EPL. Moreover, Peñas et al. [15] also found that high-intensity running by FWs in the past eight seasons of La Liga increased by 8%. These findings indicate a notable trend of enhanced physical performance among FWs in these top-tier soccer leagues over the specified periods. From our research, we found that some of our results align with previous studies [15, 17]. Specifically, we observed non-linear increasing trends in high-speed running, sprinting, and the number of sprints among FWs in the 4-1-4-1, 4-4-2, 3-4-3, and 3-5-2 formations from the 2015 to 2021 seasons. According to Bush et al. [17], an increase in high-intensity running in the EPL was caused by evolving game patterns rather than natural variability. Our research findings support this viewpoint. As the game evolves, defense becomes more organized and compact [49], consequently leading to players occupying an attacking role (such as FWs) needing to pose a direct threat to the opposition by covering a greater proportion of their distance at higher intensities. However, we observed that FWs in the 4-2-3-1 formation showed a nonlinear decrease in high-speed running but a nonlinear increase in sprinting and the number of sprints. This can be attributed to the specific tactical roles assigned to FWs in different formations. Specifically, in the 4-2-3-1 formation, there is only one FW who typically acts like a center-forward (known as a 'false nine') [20]. This FW usually receives the ball between the two opposing CDs, acting as the sole pivot point for their team's attack, which may result in FWs engaging in fewer high-speed runs. Furthermore, as the game evolves, teams are placing more emphasis on high-pressing, demanding FWs to press opponents more frequently. This could be a potential reason for the increased sprinting and number of sprints by the FWs in the 4-2-3-1 formation.

## 5. Conclusion

This study examined the non-linear evolution trends of position-specific physical and technical performance across different formations in the CSL from 2015 to 2021. The key findings

are: CDs showed nonlinear increasing trends in total distance in the 3-5-2 formation and passes in all the formations; in the 3-4-3, 4-1-4-1, and 3-5-2 formations, FBs exhibited non-linear upward trends in number of sprints and sprint distance; CMs in the 3-5-2 and 3-4-3 formations demonstrated a nonlinear downward trend in ball retention percentage but an upward trend in total distance covered; the high-speed running performed by FWs in the 4-2-3-1 formation exhibited a nonlinear increasing trend, but their counterparts in the 4-4-2 formation showed a nonlinear decline trend; WMs in the 4-4-2 formation showed a non-linear decreasing trend in ball retention percentage and passes. These results reveal the differential nonlinear evolution patterns of performance metrics dependent on playing positions and specific formations. Further research should continue monitoring these trends and investigating the underpinning factors to optimize training strategies.

## 6. Limitations

It is important to acknowledge certain limitations within this study. Although generalized additive models (GAMs) provide robust insights into non-linear trends, the results may still be influenced by external factors such as player fitness, match intensity, and weather conditions that were not controlled in this study. Additionally, the exclusion of goalkeeper data and matches with red cards or formation changes might limit the generalizability of our findings across all match scenarios in the Chinese Super League. Lastly, while we have extensive data across several seasons, the evolving nature of sports analytics and data collection might mean that newer methods could provide different insights that were not available at the time of this study.

## 7. Practical implications

The findings from this study have significant practical implications for coaching and tactical strategy in soccer. By understanding the non-linear evolution of player performance across different tactical formations, coaches can tailor training programs more effectively to meet the demands of specific positions and match situations. For example, the increased high-speed running and sprint distances identified for forwards in certain formations can inform more focused conditioning work to enhance players' endurance and speed. Additionally, the variations in technical performance, such as passing and ball retention across different formations, highlight the need for adaptable tactical training that prepares players for various strategic roles. This study thereby supports the development of targeted strategies that can optimize team performance and adaptability in competitive matches.

## Supporting information

**S1 File. The file "Datta.zip" contains the supporting information for the manuscript "non-linear trends using the generalized additive models in the physical and technical performances of the Chinese Super League players."** It includes detailed datasets used in the analysis of non-linear trends in physical and technical performance indicators of players across different positions and formations in the Chinese Super League from the 2015 to 2021 seasons. (ZIP)

## Author Contributions

**Data curation:** Rancheng Tao, Changjing Zhou, Bo Gong.

**Formal analysis:** Bo Gong.

**Investigation:** Wei Zhang.

**Methodology:** Wei Zhang, Fei Zhou.

**Writing – original draft:** Wei Zhang.

**Writing – review & editing:** Changjing Zhou.

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
