## [Decision Letter · Decision Letter 0]

23 Apr 2024

PONE-D-23-44027Non-linear trends using the generalized additive models in physical and technical performances of the Chinese Super League playersPLOS ONE

Dear Dr. Zhou,

Thank you for submitting your manuscript to PLOS ONE. After careful consideration, we feel that it has merit but does not fully meet PLOS ONE’s publication criteria as it currently stands. Therefore, we invite you to submit a revised version of the manuscript that addresses the points raised during the review process.

We look forward to receiving your revised manuscript.

Kind regards,

Julio Alejandro Henriques Castro da Costa

Academic Editor

PLOS ONE

Journal Requirements:

2. In the online submission form, you indicated that the datasets generated for this work can be accessed by contacting the corresponding author upon request.

Reviewers' comments:

Reviewer's Responses to Questions

**Comments to the Author**

1. Is the manuscript technically sound, and do the data support the conclusions?

Reviewer #1: Yes

Reviewer #2: Yes

2. Has the statistical analysis been performed appropriately and rigorously? 

Reviewer #1: Yes

Reviewer #2: Yes

3. Have the authors made all data underlying the findings in their manuscript fully available?

Reviewer #1: Yes

Reviewer #2: Yes

4. Is the manuscript presented in an intelligible fashion and written in standard English?

Reviewer #1: Yes

Reviewer #2: Yes

5. Review Comments to the Author

Reviewer #1: Dear Editor-in-Chief and Authors of the study,

I appreciate the opportunity to review the paper titled "Non-linear trends using generalized additive models in physical and technical performances of the Chinese Super League players." In addition to the methodological rigor, I commend the robustness of the analysis carried out by considering different positions, various indicators, and their nonlinear behavior over time. Below, I offer a few suggestions for the article.

L.88 – The acronym CSL has not been introduced previously. Please present the full name before using the acronym.

L.113 to 119 – If possible, present this information in table format.

Results - The figures appear to have very low quality, making it difficult to identify the variables. Therefore, we are left to "believe" in the values mentioned by the authors throughout the text. Since these figures represent the central results of the study, I suggest that the authors pay particular attention to them.

Discussion – The study by Aquino et al. (2019) could also aid in discussing the differences found between teams playing with different playing systems. "Comparisons of ball possession, match running performance, player prominence, and team network properties according to match outcome and playing formation during the 2018 FIFA World Cup."

Before finalizing the study, I suggest developing a paragraph presenting the limitations of the present work, as well as indications for future studies.

Reviewer #2: Minor revisions.

Introduction

1. In line 39, please use current references.

2. In line 42, change Barreira et al

3. In paragraph 2, any reference from the Asian Football Confederation leagues?

4. In line 88, add ‘Chinese Super League (CSL)’

Methods

1. In line 137, does the author analyse the physical and technical indicators of players in different positions?

Discussion

1. Delete line 205-209, Repetitive

2. In the discussion, it will be good to compare your findings with other AFC league teams.

3. In line 229 and throughout the manuscript, change the word ‘more cleanly’ with other appropriate words.

4. Add the limitation of the study and practical implication paragraph.

6. PLOS authors have the option to publish the peer review history of their article (what does this mean?). If published, this will include your full peer review and any attached files.

Reviewer #1: No

Reviewer #2: **Yes: **RAJA MOHAMMED FIRHAD RAJA AZIDIN

---

## [Author Response · Author response to Decision Letter 0]

26 Jun 2024

Response to Reviewer 1 comments

Dear reviewer,

Thank you for your time and valuable comments. We have provided a point-by-point response to each of your comments and suggestions and have made the appropriate changes to the manuscript. We believe the paper has improved significantly because of the review process. 

Comments 1：L.88 – The acronym CSL has not been introduced previously. Please present the full name before using the acronym.

Response: Thank you for your valuable feedback regarding the use of the acronym "CSL" in our manuscript. We have carefully considered your suggestion and have made the necessary adjustments. Specifically, we have introduced the full name "Chinese Super League (CSL)" at its first mention in lines 67–70, as follows:

"Similarly, research by Zhou et al. on the Chinese Super League (CSL) reveals that from 2012 to 2017, there was a notable enhancement in high-intensity play and offensive actions, illustrating an increase in both physical and technical demands across different playing positions."

We appreciate your attention to detail and believe this change enhances the clarity and readability of our manuscript.

Comments 2：L.113 to 119 – If possible, present this information in table format.

Response: Thank you for your insightful suggestion regarding the presentation of information in lines 113 to 119. We have taken your advice and transformed this section into a table format to enhance clarity and readability. The table has been added in lines 629-632 of the manuscript.

We believe this modification improves the overall presentation of the data and appreciate your guidance.

Comments 3：Results - The figures appear to have very low quality, making it difficult to identify the variables. Therefore, we are left to "believe" in the values mentioned by the authors throughout the text. Since these figures represent the central results of the study, I suggest that the authors pay particular attention to them.

Response: Thank you for your feedback on the quality of the figures in our manuscript. We understand the importance of clear and high-quality visual representations of our results. The original figures in our submission are indeed prepared at 300 DPI resolution to ensure clarity and precision.

However, it appears that the process of uploading the manuscript may have caused a compression issue, affecting the image quality. We apologize for any inconvenience this may have caused. To resolve this, we will re-upload the figures to ensure they are displayed at their intended quality.

We appreciate your understanding and your attention to detail, and we will ensure that the figures meet the highest standards for clarity in the revised submission

Comments 4：Discussion – The study by Aquino et al. (2019) could also aid in discussing the differences found between teams playing with different playing systems. "Comparisons of ball possession, match running performance, player prominence, and team network properties according to match outcome and playing formation during the 2018 FIFA World Cup."

Response: Thank you for your valuable suggestion regarding the inclusion of the study by Aquino et al. (2019) in our discussion. We have incorporated this reference into our manuscript to enhance the discussion on the differences found between teams using various playing systems. Specifically, we have added the following content in lines 287-295:

"According to our data, CMs in the 4-2-3-1 formation have shown nonlinear growth trends in the number of passes and ball retention percentage from the 2015 to 2021 seasons. This is in line with the findings of Aquino et al. [37], who observed that teams employing a 4-2-3-1 formation tended to dominate possession and displayed enhanced network properties, suggesting better team coordination and player interaction. Furthermore, it is widely acknowledged that CMs play a crucial role in coordinating offensive actions [33, 38]. Thus, this trend may also be attributed to build-up play, which allows CMs in the 4-2-3-1 formation to establish a connection between the defensive line and the forward line through passing."

We believe this addition strengthens the discussion by providing a comparative perspective and supporting our findings with relevant literature.

Thank you again for your insightful feedback.

Comments 5：Before finalizing the study, I suggest developing a paragraph presenting the limitations of the present work, as well as indications for future studies.

Response: Thank you for your insightful suggestion to include a discussion on the limitations of our study and indications for future research. We have developed a paragraph to address these aspects and have included it in lines 417-439 of the manuscript:

Limitations：It is important to acknowledge certain limitations within this study. Although generalized additive models (GAMs) provide robust insights into non-linear trends, the results may still be influenced by external factors such as player fitness, match intensity, and weather conditions that were not controlled in this study. Additionally, the exclusion of goalkeeper data and matches with red cards or formation changes might limit the generalizability of our findings across all match scenarios in the Chinese Super League. Lastly, while we have extensive data across several seasons, the evolving nature of sports analytics and data collection might mean that newer methods could provide different insights that were not available at the time of this study.

Practical Implications ：The findings from this study have significant practical implications for coaching and tactical strategy in soccer. By understanding the non-linear evolution of player performance across different tactical formations, coaches can tailor training programs more effectively to meet the demands of specific positions and match situations. For example, the increased high-speed running and sprint distances identified for forwards in certain formations can inform more focused conditioning work to enhance players' endurance and speed. Additionally, the variations in technical performance, such as passing and ball retention across different formations, highlight the need for adaptable tactical training that prepares players for various strategic roles. This study thereby supports the development of targeted strategies that can optimize team performance and adaptability in competitive matches.

We believe these additions provide a comprehensive view of our study's scope and offer valuable directions for future research.

Response to Reviewer 2 comments

Dear reviewer,

Thank you for your time and valuable comments. We have provided a point-by-point response to each of your comments and suggestions and have made the appropriate changes to the manuscript. We believe the paper has improved significantly because of the review process.

Your thoughtful insights and constructive feedback have been instrumental in refining our work, and we are grateful for your dedication to the review process. Your expertise has undoubtedly contributed to the overall quality of the manuscript.

Introduction

Comments 1：In line 39, please use current references.

Response：Thank you for your suggestion to use current references in line 39. We have updated the manuscript to include recent literature to support our statements. Specifically, we have cited the following references:

Lago-Peñas, C., et al., Evolution of physical and technical parameters in the Spanish LaLiga 2012-2019. Science and Medicine in Football, 2023. 7(1): p. 41-46.

Allen, T., et al., Running more than before? The evolution of running load demands in the English Premier League. International Journal of Sports Science & Coaching, 2024. 19(2): p. 779-787

We believe these recent references enhance the relevance and credibility of our study.

Comments 2：In line 42, change Barreira et al

Response：Thank you for your suggestion to update the reference in line 42. We have made the necessary changes in the manuscript to include the following citation:

Zhou, C., et al., Long-term influence of technical, physical performance indicators and situational variables on match outcome in male professional Chinese soccer. Journal of Sports Sciences, 2021. 39(6): p. 598-.

Comments 3：In paragraph 2, any reference from the Asian Football Confederation leagues?

Response：Thank you for your suggestion to include references from the Asian Football Confederation leagues in paragraph 2. We have addressed this by incorporating the following reference in lines 67-70:

"Similarly, research by Zhou et al. [16] on the Chinese Super League (CSL) reveals that from 2012 to 2017, there was a notable enhancement in high-intensity play and offensive actions, illustrating an increase in both physical and technical demands across different playing positions."

We believe this addition enriches the context of our study by providing relevant insights from the Asian Football Confederation leagues.

Comments 4：In line 88, add ‘Chinese Super League (CSL)’

Response：Thank you for your suggestion to include the full name "Chinese Super League (CSL)" in line 88. We have made this adjustment at the first mention of the league in lines 67-70, as follows:

"Similarly, research by Zhou et al. [16] on the Chinese Super League (CSL) reveals that from 2012 to 2017, there was a notable enhancement in high-intensity play and offensive actions, illustrating an increase in both physical and technical demands across different playing positions."

Methods

Comments 5: In line 137, does the author analyse the physical and technical indicators of players in different positions?

Response：Thank you for your insightful question regarding line 137. Our study indeed analyzes the physical and technical indicators of players in different positions. Specifically, we have examined the non-linear evolution of these indicators for central defenders, fullbacks, central midfielders, wide midfielders, and forwards across various formations from the 2015 to 2021 seasons in the Chinese Super League (CSL).

We employed generalized additive models (GAMs) to analyze these trends, focusing on metrics such as total distance covered, sprint distance, number of sprints, high-speed running, ball retention percentage, pass success rate, and more. These analyses provide a comprehensive understanding of how physical and technical performances have evolved position-specifically under different tactical formations.

We hope this clarifies our approach and analysis.

Thank you for your valuable feedback.

Discussion

Comments 6: Delete line 205-209, Repetitive

Response: Thank you for your careful review of our manuscript. We have addressed your suggestion and deleted the repetitive content in lines 205-209.

We appreciate your attention to detail and believe this change enhances the clarity of our manuscript.

Comments 7: In the discussion, it will be good to compare your findings with other AFC league teams.

 Response: Thank you for your suggestion to compare our findings with other AFC league teams in the discussion. Due to the limited research available on the evolution of running and technical performance trends of players in Asian leagues, we have made a comparison with the Second Division of Japanese Professional Football (J2 League) in lines 235-237 of our manuscript:

"Previous research on the Second Division of Japanese Professional Football indicates that the total running distance of players has shown a decreasing trend over three seasons [31]."

We recognize that this comparison is somewhat limited and appreciate any further suggestions you may have to enhance this section of our manuscript.

Thank you for your valuable feedback and understanding.

Comments 7: In line 229 and throughout the manuscript, change the word ‘more cleanly’ with other appropriate words.

 Response: Thank you for your suggestion to replace the phrase "more cleanly" with more appropriate words throughout the manuscript. We have made the necessary changes in lines 228-229 and 270-272 as follows:

"These results may suggest that players play soccer with more tactical discipline and less aggressiveness [29, 30]."

"The diminishing challenge faced by FBs, similar to what has been observed among CDs, might indicate a shift towards a less aggressive style of play among players [29, 30]."

We believe these revisions improve the clarity and precision of our language.

Comments 7: Add the limitation of the study and practical implication paragraph

Response: Thank you for your insightful suggestion to include a discussion on the limitations of our study and indications for future research. We have developed a paragraph to address these aspects and have included it in lines 417-439 of the manuscript:

Limitations：It is important to acknowledge certain limitations within this study. Although generalized additive models (GAMs) provide robust insights into non-linear trends, the results may still be influenced by external factors such as player fitness, match intensity, and weather conditions that were not controlled in this study. Additionally, the exclusion of goalkeeper data and matches with red cards or formation changes might limit the generalizability of our findings across all match scenarios in the Chinese Super League. Lastly, while we have extensive data across several seasons, the evolving nature of sports analytics and data collection might mean that newer methods could provide different insights that were not available at the time of this study.

Practical Implications ：The findings from this study have significant practical implications for coaching and tactical strategy in soccer. By understanding the non-linear evolution of player performance across different tactical formations, coaches can tailor training programs more effectively to meet the demands of specific positions and match situations. For example, the increased high-speed running and sprint distances identified for forwards in certain formations can inform more focused conditioning work to enhance players' endurance and speed. Additionally, the variations in technical performance, such as passing and ball retention across different formations, highlight the need for adaptable tactical training that prepares players for various strategic roles. This study thereby supports the development of targeted strategies that can optimize team performance and adaptability in competitive matches.

We believe these additions provide a comprehensive view of our study's scope and offer valuable directions for future research.

---

## [Decision Letter · Decision Letter 1]

12 Jul 2024

Non-linear trends using the generalized additive models in physical and technical performances of the Chinese Super League players

PONE-D-23-44027R1

Dear Dr. Zhou,

We’re pleased to inform you that your manuscript has been judged scientifically suitable for publication and will be formally accepted for publication once it meets all outstanding technical requirements.

Kind regards,

Julio Alejandro Henriques Castro da Costa

Academic Editor

PLOS ONE

Additional Editor Comments (optional):

Reviewers' comments:

Reviewer's Responses to Questions

**Comments to the Author**

1. If the authors have adequately addressed your comments raised in a previous round of review and you feel that this manuscript is now acceptable for publication, you may indicate that here to bypass the “Comments to the Author” section, enter your conflict of interest statement in the “Confidential to Editor” section, and submit your "Accept" recommendation.

Reviewer #2: All comments have been addressed

2. Is the manuscript technically sound, and do the data support the conclusions?

Reviewer #2: Yes

3. Has the statistical analysis been performed appropriately and rigorously? 

Reviewer #2: Yes

4. Have the authors made all data underlying the findings in their manuscript fully available?

Reviewer #2: Yes

5. Is the manuscript presented in an intelligible fashion and written in standard English?

Reviewer #2: Yes

6. Review Comments to the Author

Reviewer #2: All required comments, questions have been answered and that all responses meet expectations and, formatting specifications.

7. PLOS authors have the option to publish the peer review history of their article (what does this mean?). If published, this will include your full peer review and any attached files.

Reviewer #2: **Yes: **RAJA MOHAMMED FIRHAD BIN RAJA AZIDIN

---

## [Editor Report · Acceptance letter]

17 Jul 2024

PONE-D-23-44027R1 

PLOS ONE

Dear Dr. Zhou, 

I'm pleased to inform you that your manuscript has been deemed suitable for publication in PLOS ONE. Congratulations! Your manuscript is now being handed over to our production team.

Kind regards, 

on behalf of

Dr. Julio Alejandro Henriques Castro da Costa 

Academic Editor

PLOS ONE